# A Probability-Based Models Ranking Approach: An Alternative Method of Machine-Learning Model Performance Assessment

**DOI:** 10.3390/s22176361

**Published:** 2022-08-24

**Authors:** Stanisław Gajda, Marcin Chlebus

**Affiliations:** Faculty of Economic Sciences, University of Warsaw, Długa Street 44/50, 00-241 Warsaw, Poland

**Keywords:** machine learning, model performance assessment, model selection, hyperparameters tuning, model performance measures, Elo-based Predictive Power, mixed effects logistic regression

## Abstract

Performance measures are crucial in selecting the best machine learning model for a given problem. Estimating classical model performance measures by subsampling methods like bagging or cross-validation has several weaknesses. The most important ones are the inability to test the significance of the difference, and the lack of interpretability. Recently proposed Elo-based Predictive Power (EPP)—a meta-measure of machine learning model performance, is an attempt to address these weaknesses. However, the EPP is based on wrong assumptions, so its estimates may not be correct. This paper introduces the Probability-based Ranking Model Approach (PMRA), which is a modified EPP approach with a correction that makes its estimates more reliable. PMRA is based on the calculation of the probability that one model achieves a better result than another one, using the Mixed Effects Logistic Regression model. The empirical analysis was carried out on a real mortgage credits dataset. The analysis included a comparison of how the PMRA and state-of-the-art k-fold cross-validation ranked the 49 machine learning models, an example application of a novel method in hyperparameters tuning problem, and a comparison of PMRA and EPP indications. PMRA gives the opportunity to compare a newly developed algorithm to state-of-the-art algorithms based on statistical criteria. It is the solution to select the best hyperparameters configuration and to formulate criteria for the continuation of the hyperparameters space search.

## 1. Introduction and Literature Overview

Model performance assessment plays a crucial role in machine learning. The applied comparison procedure matters during model selection for a given problem, in hyperparameters tuning, and while testing the newly proposed algorithm. Depending on the characteristic of the problem and expected properties, different measures are applied [1]. Among those most commonly used in recent years are: Accuracy [2], F1 [3], cross-entropy [4], AUC [5] for classification and RMSE [6], MSE [7], MAE [8] for regression. The classic approach divides available datasets into train, validation, and test sets. A disadvantage of this method is that obtained measure estimate has a high variance. In order to reduce it, subsampling methods are applied. Cross-validation and bootstrap aggregating (bagging) are most often used. In [9], it is claimed that many commonly used subsampling schemes suffer from substantial negative bias when considering AUC in the small-sample setting. The variance of the obtained results is also an important issue. In [10], it is shown that there exists no universal (valid under all distributions) estimator of the variance of cross-validation.

In [11], the authors described the two most often used approaches to calculate AUC and different measures over folds: *pooling* and *averaging*. In pooling, the results obtained in each fold are joined into one set on which the AUC measure is calculated. Averaging assumes the calculation of the mean value of AUC measures obtained in all folds. Parker et al. show in their study on low-signal simulated data and Van’t Veer breast cancer dataset that pooling may cause substantial biases due to comparing observations from different test samples [12]. They explain why using k-fold cross-validation modifications: *Balanced cv*, *Stratified cv*, *Balanced leave-one-out cv* reduce the bias for small datasets. In [13], Varoquaux states that the use of cross-validation on small sample datasets leads to large error bars. The simulation that was carried out in a biological context on small sample data showed that the cross-validation approach provides inaccurate results—with, e.g., a 10% error bar for accuracy measure on a sample with 100 observations. In [14], Krzanowski proposed an approach that combines the advantages of pooling and averaging—*Leave-pair-out cross-validation* (LPOCV). It involves calculating AUC values for each positive-negative pair. The final AUC value is obtained by averaging through all of these pairs. Efficient LPOCV implementation was developed by Pahikkala et al. [15]. The *Leave-one-out cross-validation* (LOOCV) is equivalent to LPO for measures other than AUC The limitation of both algorithms is their complexity—the required number of training rounds is of the order: $O(n^{2})$, O(n) for LPOCV and LOOCV respectively, where n is a number of observations. Therefore these approaches may be applied only to small datasets. LPOCV and LOOCV, similarly to pooling, make maximal use of the available dataset. However, these methods are not perfect—in [16], Gronau et al. show their limitations.

An essential aspect of the model selection process is testing the significance of the performance difference between two models. In [17], the authors conducted an extensive review of the statistical methods used for model selection. They described several methods to compare algorithms, including fitting and evaluating models via cross-validation. The first of the presented methods to test performance difference significance is the difference of the proportions test. This involves comparing confidence intervals of two accuracies using a z-score. However, Dietterich in his simulation study states that this test tends to have a high false positive rate [18]. Instead, Dietterich recommends using McNemar’s test [19]. The test statistic is calculated from the number of observations for which models’ predictions were different. *p*-values are calculated based on the Chi-square distribution. When the number of differences in predictions is relatively small, the Chi-squared distribution may be imprecise. In this case, the binomial test is recommended.

Methods mentioned above: *Difference of proportions test*, *McNemar’s test* and *Binomial test* all assume specific threshold level, which determines the models’ confusion matrix. What is more, these methods compare two models. Comparison of multiple models’ performance can be made using *Cochran’s Q test*. It verifies the null hypothesis that accuracy of *n* models is equal, but it does not indicate which models differ [20].

Resampling-based testing procedures are also of practical importance for the model selection problem. One of them is Dietterich’s *5 × 2-Fold Cross-Validated Paired t-Test* that assumes performing 5 iterations of 2-fold cross-validation [18]. The more robust alternative: *Combined 5 × 2 cv F-test* was proposed by Alpaydm [21]. In [22], the authors presented machine learning models comparison methods, which are most commonly used in practice. Beyond the above-mentioned *5 × 2-Fold Cross-Validation Paired t-Test* and *McNemar’s test* there is *paired Student’s t-test* described with further correction proposed in [23]. Although the approaches discussed above give accurate results, they are not often used. They are an addition to the classical approach of averaging and are not directly embedded in it.

Elo-based Predictive Power (EPP)—a measure introduced in [24] is the answer to these drawbacks of classical approaches to model performance assessment. It assumes that the performance of every pair of models is compared. The outcome of the comparison is the input to the logit model. The logit model coefficients indicate the predictive ability of the analyzed models. This approach is novel as it considers all models at once in comparison. Furthermore, the estimated values of the measure are interpretable and it is possible to use a statistical test to investigate the significance of the difference between two models’ performance. However, the EPP does not consider the lack of independence of comparisons made on the same subsample. Therefore, its estimates are based on fundamentally flawed assumptions, underestimating results variance, which makes its indications potentially unreliable. EPP aggregates relative information about models’ pairs comparison and turns it into one value per model describing their predictive power. The different approach to decision making in context of subjective information is presented in [25,26].

The aim of this article is to introduce the Probability-based Models Ranking Approach (PMRA), a method of evaluating the performance of a machine learning model, which makes it possible to interpret the difference in the performance of 2 models and to test the statistical significance of this difference, the method is also more reliable than the state-of-the-art averaging approach. PMRA is a modification of the EPP that retains its advantages over the state-of-the-art approach and, at the same time, addresses its main disadvantage. Using a mixed effects model gives a more realistic estimate of the variance, which allows greater confidence in indications of statistical tests stating whether two models’ performances are significantly different or not. In the study, PMRA is practically applied to the results of various machine learning models obtained on the Multifamily mortgages dataset describing real credits acquired by Fannie Mae in the years 2000–2020.

The paper is organized as follows: in the next section, the main limitations of the subsampling approach to model performance assessment are described. In Section 3, EPP measure construction, based on the logit model, is presented with its weaknesses. Then conception of the Probability-based Models Ranking Approach (PMRA) using mixed effects logistic regression is introduced. There is a discussion of statistical tests that can be used in the PMRA—Wald and Likelihood-ratio tests. Then we described possible practical applications of this method in hyperparameters tuning and in models’ ranking creation. Section 4 contains empirical analysis. It begins with the used dataset description, then PMRA and state-of-the-art averaging approaches are compared in the most common practical applications. Finally, the EPP and PMRA approaches are compared, showing the practical advantages of PMRA over EPP.

## 2. Weaknesses of Classical Model Assessment Methods Based on Cross-Validation

The most commonly used subsampling methods of evaluating model performance are cross-validation and bootstrap aggregating (bagging). The idea behind both of them is to generate from whole dataset (X,Y)*n* subsamples $\left(X_{1},Y_{1}\right),\ldots ,\left(X_{n},Y_{n}\right)$. Cross-validation assumes dividing (X,Y) into n equal parts and placing in each subsample all of them except one [27]. Bagging involves generating new training datasets, by sampling from (X,Y) uniformly and with replacement (some observation may be chosen multiple times) [28]. Then the algorithm is trained on each dataset and tested on observations not belonging to it. Values $r_{1},\ldots ,r_{n}$ of one of classical measures (e.g., AUC, $F_{1}$ for classification, MSE, MAE for regression) are obtained and then averaged.
(1)$$\bar{r}=\frac{r_{1}+\ldots +r_{n}}{n}$$

The resulting value $\bar{r}$ stands for model performance over a dataset. Inequality between two models’ average results is the basis of the statement that one model can learn dependencies between independent and dependent variables better than the other one [29]. In this section, we will present four main weaknesses of using this classical approach for the AUC measure example.

### 2.1. Lack of Interpretability

Most commonly used performance measures are interpretable. For example, AUC represents the probability that a model assigned a higher probability of belonging to a positive class to a randomly chosen positive example than to a randomly chosen negative example [30]. Therefore, the difference between the two models’ performances may be interpreted. However, for many classic measures, the subsampling approach, assuming averaging measure values obtained in specific samples, suffers from biases [31]. This problem is particularly relevant for small sample problems [9]. The reasons mentioned make it impossible to rely on the interpretation of the results.

### 2.2. Lack of Statistical Tools to Measure the Statistical Significance of Results Differences

For many performance measures, it is possible to test the significance of the difference between two models’ performances. However, making conclusions about the significance of performance difference based only on the obtained average value of this measure is a more complex issue. For example for AUC, there are statistical tests to conclude about equality of two AUC results [32,33]. However, their use to compare mean AUC would be incorrect as they are based on ROC curve analysis. As described in the literature overview—the most commonly used methods (Paired *t*-test, *5* × *2 folds cv*) are imperfect due to their pairwise character.

### 2.3. Sensitivity on Outliers

Conclusions based on cross-validation average AUC can also be misleading when one of the model’s obtained AUC scores stands out greatly from the others. It may result in having such a strong impact on mean AUC that it does not reflect the expected relation between the two models’ results. The model which performs worse in most cases may be considered better because of just one outlying observation. Figure 1 presents results obtained by 2 models in 5-fold cross-validation. M1 model’s results are better in 4 of 5 folds. The extraordinarily high result of model M2 on the fourth fold overstates its mean AUC so that it is higher than the mean AUC of M1. Similarly, in the cross-validation, presented in Figure 2, the surprisingly low score of model M1 in the fourth fold underscores its mean AUC so that it is lower than the mean AUC of the M2 model.

### 2.4. Ignoring in Analysis Variance of Outcomes in Specific Folds

Measuring the model’s performance using cross-validation assumes simple averaging of obtained scores from all test folds—the result from each fold is taken with equal weight. However, distributions of models’ results may differ between folds. Variance in one fold may be several times higher than in the other. The omission of this factor in the analysis may distort the results. Figure 3 presents the diversity of 49 models AUC results among 10 folds. Results were calculated on the credit-risk dataset. The highest standard deviation (8th fold) is over 3 times higher than the lowest (5th fold).

Not taking diversity of other models’ performances in the fold may lead to a paradoxical situation, where 2 models ranked similarly get considerably different mean AUC values, depending on whether the folds where they scored well had high or low variance. Quantile and AUC results of 2 models presented in Table 1 illustrate this situation. In 2 folds of cross-validation M1 and M2 placed in 60 and 75 percentile. M1 and M2 performed the same, comparing their results to other models. However, their average AUC values differ. The reason for that is different variance in 1st and 2nd fold.

Listed weaknesses of classical subsampling performance assessment methods mean one cannot be sure of their reliability in ordering models. Moreover, these methods do not provide sufficient statistical tools to distinguish between models with better and those with poorer performance in practical applications. The answer to the described challenges are measures based on the probability of win—EPP and PMRA presented in another section.

## 3. Probability Measures of Model Performance

In order to resolve classic subsampling approach weaknesses mentioned in Section 2, it is recommended to use measures based on the estimation of the probability that one model will achieve better results than another. This section begins by introducing Elo-based Predictive Power measure proposed in [24], which initiated a probability-based approach to the model performance assessment. Further, it is shown how the EPP measure addresses four disadvantages of classic performance measures listed in Section 2. Then Probability-based Models Ranking Approach is introduced, which is the EPP extension resolving its crucial weaknesses. The section concludes with a description of the possible applications of probability model performance measures.

### 3.1. Elo-Based Predictive Power (EPP)

Assume $M=\{M_{1},\ldots ,M_{m}\}$ is a set of machine learning algorithms with fixed hyperparameters, which were evaluated in k-fold cross-validation. The EPP approach is to assign them scores $\beta_{1},\ldots ,\beta_{m}$ representing their predictive power. The higher score, the better model is assessed. This measure is constructed so that difference between 2 models’ scores is interpretable. From value of $\beta_{i}-\beta_{j}$ difference of $M_{i},M_{j}$ models’ scores probability that model $M_{i}$ performance expressed in classic measure value beats $M_{j}$ performance, can be calculated. EPP scores can be estimated as coefficients of a following logistic regression model with intercept $\beta_{0}=0$.
(2)$$\ln\left(\frac{p_{i,j}}{1-p_{i,j}}\right)=\beta_{1}x_{1}+\beta_{2}x_{2}+\ldots +\beta_{m}x_{m}$$
where:

xa=1a=i−1a=j;

$p_{i,j}$—probability, that model $M_{i}$ gets better result than model $M_{j}$.

Logistic Regression Model is estimated on cross-validation results dataset transformed, so that it reflects effect of comparision of every pair of models. Details about dataset transformation are described in Section 3.3 From the model’s formula we may obtain the exact probability value:(3)$$p_{i,j}=\frac{e^{\beta_{i}-\beta_{j}}}{e^{\beta_{i}-\beta_{j}}+1}.$$

Such constructed EPP score addresses four classic model performance measures weaknesses listed in Section 2.

**Lack of interpretability:** EPP score is interpretable. It gives a clear answer about the probability of one model obtaining better results than another.**Lack of statistical tools to measure statistical significance of results differences:** EPP scores are logistic regression coefficients estimates. To check if two models’ performance difference is significant we can use Wald or Likelihood Ratio tests. They are able to verify linear null hypotheses like: $H_{0}:\beta_{i}-\beta_{j}=0$, which implies $p_{i,j}=\frac{1}{2}$.**Sensitivity on outliers:** The outstanding result of the model in one of the folds, of course, will positively impact the assessment of its performance. Nevertheless, the decision will be based on the relation observed in the majority of folds.**Omitting in analysis variance differences between folds:** Data from every fold have an equal impact on fixed effects estimation. The key impact on the EPP score has not the exact value of the measure, but the order of results, those models obtained in specific folds.

The EPP approach is innovative and appears to improve our ability to analyze the model’s performance. However, it has one major drawback. One of the logistic regression assumptions is the independence of observations. This assumption is not satisfied because all $\frac{n(n-1)}{2}$ observations expressing models’ comparisons made in one fold are not independent. As a result, variance-covariance matrix estimates may be inconsistent [34], which results in underestimated *p*-value in Wald and *LR* tests. This may cause one model’s performance to be mistakenly considered better than another. As a result, the use of specific algorithms may be rejected, even though they are not significantly weaker.

### 3.2. Probability-Based Models Ranking Approach

As a solution to the main EPP problem, we propose a Probability-based Models Ranking Approach, which is a modification of the EPP approach. The probability of model $M_{i}$ overcoming model $M_{j}$(i<j) is estimated using generalized mixed effects model with logit as a link function in version defined by formula [35]:(4)$$ln\left(\frac{p_{i,j}}{1-p_{i,j}}|fold=k\right)=\beta_{0}+\sum_{s=1}^{n}\beta_{s}*X_{s}+u_{k}+\epsilon$$
where:

xa=1a=i−1a=j;

$p_{i,j}$—probability, that model $M_{i}$ gets better result than model $M_{j}$;

$\beta_{0},\ldots ,\beta_{n}$—fixed effects coefficients;

$u_{k}$—random intercept depending on number of fold;

ϵ—random error.
$$u_{1},\ldots ,u_{k}∼N\left(0,\;\Sigma_{intercept}^{2}\right)\;.$$$$\epsilon ∼N(0,\;\sigma^{2})\;.$$

From such a formulated model, we can easily calculate probabilities of results between models $M_{i}$ and $M_{j}$. We calculate the expected value of both sides of the formula.
(5)$$\exp\left(\ln\left(\frac{p_{i,j}}{1-p_{i,j}}|fold=k\right)\right)=\exp\left(\beta_{0}+\sum_{s=1}^{n}\beta_{s}*X_{s}+u_{k}+\epsilon \right)$$
(6)$$\ln\left(\frac{p_{i,j}}{1-p_{i,j}}\right)=\beta_{0}+\beta_{i}-\beta_{j}$$
(7)$$p_{i,j}=\frac{\exp(\beta_{0}+\beta_{i}-\beta_{j})}{1+\exp(\beta_{0}+\beta_{i}-\beta_{j})}$$

#### 3.2.1. Statistical Tests

Two statistical tests may be helpful to verify linear hypotheses on the values of fixed effects—Likelihood Ratio Test (LR test) and Wald Test. They both allow us to test whether a set of s independent linear hypotheses is true. The null and alternative hypotheses for both of them are:(8)$$H_{0}:K\beta =c$$
(9)$$H_{1}:K\beta \neq c$$
where:

*K*—known, full rank matrix, rank(*K*) = s;

β—vector of fixed effects.

The Likelihood Ratio Test (LRT) compares two nested models—general and restricted. They are built on the same data, but some fixed effects are omitted in the restricted one. Therefore, the model without the considered effect is a special case of the model with this effect [36]. Test may be applied when both models were fitted using maximum likelihood estimation. The *LR* test statistic is as follows [37]:(10)$$LR=-2(l\left(\tilde{\beta },\tilde{\Sigma }|y\right)-l\left(\hat{\beta },\hat{\Sigma }|y\right))$$
where $\left(\tilde{\beta },\tilde{\Sigma }\right)$ are estimations of fixed effects and their variance-covariance matrix in restricted model. $\left(\hat{\beta },\hat{\Sigma }\right)$ are estimations in the general model.

The Wald test statistic *w* given in [37]:(11)$${\left[K\hat{\beta }-c\right]}^{T}\left[KF_{\hat{\beta }}^{-1}K^{T}\right]\left[K\hat{\beta }-c\right]$$

$F_{\hat{\beta }}$ can be either observed and expected information matrix evaluated in $\hat{\beta }$:(12)$$F_{\hat{\beta }}=-\frac{\partial^{2}l\left(\hat{\beta },\hat{\Sigma }|y\right)}{\partial \beta \partial \beta^{T}}{|}_{\beta =\hat{\beta }}$$
or
(13)$$F_{\hat{\beta }}=E\left(-\frac{\partial^{2}l\left(\hat{\beta },\hat{\Sigma }|y\right)}{\partial \beta \partial \beta^{T}}{|}_{\beta =\hat{\beta }}\right)$$

Both tests statistics *w* and LR are chi-square distributed with rank(K) degrees of freedom. The Wald test is a quadratic approximation to the log-likelihood function by a second-order Taylor expansion and is an approximation to the *LR* statistic. *p*-values obtained for the *LR* test are more exact than the *p*-values obtained for the Wald test. The disadvantage of the *LR* test is that both the restricted and the unrestricted models have to be fitted, whereas, for the Wald test, only the unrestricted model is needed. With increasing sample size, the log-likelihood becomes approximately quadratic so that both tests are asymptotically equivalent [37].

In the Probability-based Models Ranking Approach, either Wald or *LR* tests can be applied to test hypotheses of two kinds. The first one has the following null and alternative hypotheses:$$H_{0}:\beta_{0}+\beta_{i}-\beta j=0$$
$$H_{1}:\beta_{0}+\beta_{i}-\beta j\neq 0$$
what is equivalent to:$$H_{0}:p_{i,j}=\frac{1}{2}$$
$$H_{1}:p_{i,j}\neq \frac{1}{2}$$

It allows us to verify hypotheses about the equality of two models’ performances. Another application of *LR* and Wald tests in PMRA includes testing the following null and alternative hypotheses:$$H_{0}:\beta_{i_{1}}=\ldots =\beta_{i_{l}}=0$$
$$H_{1}:\exists k\in \left\{1,\ldots ,l\right\}:\beta_{k}\neq 0$$

It allows us to test whether a group of fixed effects is significant or can be deleted from the model. Removing the insignificant fixed effect corresponding to a given model from the mixed effects model causes a slight change in the estimate of the performance of that model. In return, reducing the complexity improves the precision of estimates of fixed effects and variance for the best models (which estimates are crucial in most practical applications).

#### 3.2.2. Model Estimation

The described mixed effects model may be estimated using the lme4 package [38]. Coefficients are estimated with maximum likelihood fitting. The approximation algorithm assumes that the Penalized Least Squares method has been used multiple times.

Data matrix from transformed dataset described in Section 3.3 does not have a full columnar rank because the sum of all fixed effects columns is equal to the zero vector. Hence, it is necessary to choose $i_{zero}\in \left\{1,\ldots ,n\right\}$ for which $\beta_{i_{zero}}=0$ is assumed. This does not affect the results because the coefficients’ values are relative. Using the weakest model as model zero is recommended. Then, the subsequently applied procedure eliminating insignificant fixed effect has no negative impact on the estimates of the best models. After the first model estimation, fixed effects, whose coefficients are statistically equal to zero, should be eliminated. There are several methods of choosing a set of insignificant variables. They differ in their accuracy and computation time. The algorithms below are ordered from the slowest but most accurate to the fastest but least accurate.


**Rigorous backward approach**
(a)sort fixed effects of current model by coefficients’ *p*-value obtained in Wald test ($H_{0}:\beta_{i}=0$) in descending order.(b)try to eliminate any of fixed effects from current model, beginning from these with highest *p*-value. Use Likelihood Ratio Test to test joint significance of fixed effects: $X_{i_{1}},\ldots ,X_{i_{l}},X_{k}$, where $X_{i_{1}},\ldots ,X_{i_{l}}$ are already eliminated fixed effects and $X_{k}$ is currently inspected fixed effect $\left(H_{0}:\beta_{i_{1}}=\ldots =\beta_{i_{l}}=\beta_{k}=0\right)$(c)if any fixed effect was eliminated get back to point (a) with restricted model, finish otherwise.**Accelerated backward approach**: The same as Rigorous backward approach, but in point (b), consider eliminating only fixed effects whose *p*-value is above a certain assumed confidence level α.**Heuristic backward approach**: Until we get the model for which all coefficients are significant, eliminate the fixed effect with the highest Wald Test *p*-value.

The Accelerated backward approach seems to be a good compromise between speed and accuracy.

### 3.3. Data Preparation

Data on which probability measures of model performance are based is the result of a cross-validation process with itemized test folds numbers on which assessment was made. The form of the data is as follows:(14)(Model,Fold,AUC)

Example input dataset is presented in Table 2. In the preparation process, the form of the data must be changed to fit the EPP and PMRA models. The new form does not contain exact information about the AUC value. It only provides information for each pair of models, which one performed better (has a higher AUC value). If in k-th fold models $\left(M_{i},M_{j}\right)$ (i<j) scored ($AUC_{M_{i},k},AUC_{M_{j},k}$) the following observation will be added to new dataset.
(15)$$\left(X_{1},X_{2},\ldots ,X_{m},Fold,Result\right)$$
where:(16)$$X_{s}\;\;\;\;=\;\;\;\;\left\{\begin{array}{cc} \;1 & \;\mathrm{if}\;s\;=\;i\\ -1 & \;\mathrm{if}\;s\;=\;j\\ \;0 & \;\mathrm{otherwise} \end{array}\right.$$
(17)$$result=\left\{\begin{array}{cc} 1 & \mathrm{if}AUC_{M_{i},k}>AUC_{M_{j},k}\\ 0 & \mathrm{otherwise} \end{array}\right.$$
(18)Fold=k

Table 3 shows example dataset from Table 2 after transformation. In first fold occurs:
$$AUC_{M_{1},1}>AUC_{M_{2},1}>AUC_{M_{3},1}$$therefore, the first 3 rows of the transformed table encode the following information:
$$AUC_{M_{1},1}>AUC_{M_{2},1}\;AUC_{M_{1},1}>AUC_{M_{3},1}\;AUC_{M_{2},1}>AUC_{M_{3},1}$$

If the input data are results of n models obtained in k-fold cross validation, then the table we have at the beginning is (m∗k)×3 and after-transformation table is $\frac{m(m-1)}{2}*k\times \left(m+2\right)$.

### 3.4. Comparision of PMRA to EPP

The Probability-based Models Ranking Approach is a modification of the innovative Elo-based Predictive Power (EPP) approach introduced in [24]. Modified model loses some of EPP’s features to address its weaknesses and improve its efficiency in some of the applications.

The main problem with EPP is that one of the logistic regression assumptions—independence of observations—is not satisfied. The results of *n* models in each fold of input dataset generates $\frac{n(n-1)}{2}$ observations expressing model comparisons. These observations are not independent. As a result, the estimated variance-covariance matrix is underestimated. This makes the EPP tend consider models that have statistically equal performances as different. An attempt to solve this problem is implementing random effects in the Probability-based Models Ranking Approach. Random effects cluster the results obtained in the same fold. As a result, the estimate of the variance-covariance matrix of fixed effects is more accurate. PMRA allows us to rely on statistical tests’ *p*-values and make decisions in the model selection process based on statistical criteria rather than arbitrarily established conditions.

Another difference is that PMRA contains the procedure of eliminating fixed effects corresponding to models statistically equal to model zero. This General to Specific approach may distort estimates of the probability of a win for models corresponding to deleted fixed effects. However, the model becomes less complex, and the coefficients’ variance decreases. PMRA enables better estimation of the probability of win and equal performance tests *p*-values for models which rated best (in most applications, we are usually more interested in proper performance evaluation of models at the top of the ranking than those from the bottom).

The different design of the Probability-based Models Ranking Approach means that, unlike EPP, it does not have the convenient property that performance can be specified by a single number. Moreover, it cannot be used to compare results from different datasets. What PMRA gives in return is a better estimation of the variance-covariance matrix resulting in better estimates of *p*-values in tests for equality of two models. This makes PMRA equality of performance of two models reliable in contrast to EPP. In summary, the PMRA is a modification of the EPP lacking its universal properties. In return, its indications are reliable, what enables using it in practical applications.

### 3.5. Probability Measures Applications

Thanks to the described advantages over EPP, PMRA is suitable for practical applications. Its interpretability and statistical properties constitute advantages over the classical subsampling approach in problems of hyperparameter tuning and creating a ranking of models.

#### 3.5.1. Application in Models Ranking Building

Result of Probability-based Models Ranking Approach estimation may be basis for creation of $\left\{M_{1},\ldots ,M_{m}\right\}$ models ranking in 2 steps.

Probability matrix calculation $\begin{array}{cc} \begin{array}{cccc} M_{1} & M_{2} & \ldots & M_{m} \end{array}\\ \left[\begin{array}{cccc} p_{1,1} & p_{1,2} & \ldots & p_{1,m}\\ p_{2,1} & p_{2,2} & \ldots & p_{2,m}\\ \ldots & \ldots & \ldots & \ldots \\ p_{m,1} & p_{m,2} & \ldots & p_{m,m} \end{array}\right] & \begin{array}{c} M_{1}\\ M_{2}\\ \ldots \\ M_{m} \end{array} \end{array}$Probabilities values can be estimated as desribed in Equation (Equation 7). In case fixed effect corresponding to i-th model was eliminated, assume $\beta_{i}=0$Assigning places 1,2,…,m in ranking to models $M_{i_{1}},M_{i_{2}},\ldots ,M_{i_{n}}$ so that:for each k,l∈{1,2,…,m}:k<l occurs: $p_{i_{k},i_{l}}>\frac{1}{2}$Theoretically, it may happen that models cannot be ranked in this way because several of them form a cycle in which each model beats the next with a probability higher than 0.5 e.g., $p_{i_{k},i_{l}}>\frac{1}{2}$, $p_{i_{k},i_{l}}>\frac{1}{2}$, $p_{i_{k},i_{l}}>\frac{1}{2}$. However, such a case rarely occurs in practice and usually means that the models are insignificantly different. Therefore in this case, these models get the same ranking number.

#### 3.5.2. Application in the Process of Hyperparameters Tuning

There are many hyperparameters tuning methods like: *grid search*, *random search*, *bayesian optimization* [39] or *gradient-based optimization* [40]. Probability-based Models Ranking Approach may provide valuable information supporting model selection decisions during these procedures. Regarding the example of the iterated grid search [41]: while tuning 2 hyperparameters α,γ whose values come from the interval [0,1] the grid Ω of m∗n potential parameters sets is created:(19)$$\Omega_{1}=A\times B$$
where A,B are sets of potential values of α,γ respectively. For each hyperparameters set a model performance is calculated using cross-validation and the best one (a,b) is chosen. After that, another grid of values in neighbourhood of best set is created and the process is repeated until we decide the result will not improve significantly by further tuning.
(20)$$\Omega_{2}=\left\{\left(a\pm \epsilon_{i},b\pm \delta_{j}\right)\right\}$$

Probability-based Models Ranking Approach may be helpful in two ways:It can indicate which hyperparameters set is the best of considered in the iteration and estimate the probability of overperforming other models by it. Using statistical tests, it can be determined which of the different hyperparameters’ sets was not significantly worse than the best and should be included in another search iteration.It can also be a basis for formulating stop conditions for a further grid search. If changes in parameters cause performance changes that are not statistically significant, a continuation of the search is pointless. For example, conditions of this kind may be used: *Stop further search if significantly better hyperparameters set was not found in last k iterations*.

## 4. Real Data Application

Probability-based Models Ranking Approach may have several applications in the selection of machine learning model. PMRA may be used to create an interpretable ranking of models solving a specific task. It allows us to check which algorithms are best suited to this particular problem. When the algorithm is chosen, PMRA may be helpful in the hyperparameters tuning process. Probability-based Models Ranking Approach gives statistical tools to select the set of optimal parameters and to set up stopping conditions, which will limit computation time to the necessary minimum. This section describes these applications using results obtained on a processed dataset containing real mortgage credit repayment history. Credits included in this dataset describe multifamily loans (5+ residential units) acquired by Fannie Mae (https://multifamily.fanniemae.com/news-insights/multifamily-wire/fannie-mae-multifamily-reports-q1-2022-financial-results, accessed date: 1 April 2021) in years 2000–2020.

### 4.1. Dataset Description

The dataset on which the probability-based models were built consisted of AUC measures of 49 different models obtained from 10-fold cross-validation. In the study, the following algorithms were used:Ada Boost (10 hyperparameters sets),K-Nearest Neighbors (9 hyperparameters sets),Decision Tree (10 hyperparameters sets),Random Forest (10 hyperparameters sets),XGBoost (10 hyperparameters sets).

These algorithms were trained and tested on the Fannie Mae Multifamily Mortgages dataset. Fannie Mae is USA Government Sponsored Enterprise operating on the secondary mortgage market. After preprocessing, the dataset counted 224,988 observations of 62 features and default flag (dependent binary variable) describing 54,160 credits acquired in the period 2000–2020. The default rate was around 1%. Data included repayment history, property information, loan repayment terms, and information about the borrower’s cash flow. Table 4 and Table 5 present 10 variables with the highest discriminatory power.

### 4.2. Application in Models Ranking Building

The crucial step in solving any problem with machine learning is selecting a model appropriate to the given problem. As described in Section 4.1, 5 algorithms, each with several hyperparameters sets were tested on a real mortgage credits dataset. Their results were analyzed using PMRA and averaging approach. The use of PMRA allowed us to obtain the probability of winning in the comparison of each pair of models, as well as statistically testing whether the models’ performances are significantly different. The estimated probabilities and Wald test *p*-values for top 10 models comparisons are presented in Figure 4 and Figure 5. Average AUC and Probability-based Models Ranking Approach rankings calculated on real data are presented in Table 6 and Table 7 respectively. The top 2 models in both rankings are the same, but in further places, differences are visible. Model RF9 is best in both rankings. Nevertheless, applying the Wald test on 5% confidence level shows that XGBoost configurations, which took places 2–4, are statistically equal to it. The estimated probabilities confirm that. The probability of an RF9 win against XGB6 is only 0.505. This suggests that both algorithms, Random Forest and XGBoost, should be considered in further analysis. It is worth noticing that three different XGBoost’s hyperparameters sets were recognized as equal to the best model. It indicates that XGBoost in this application will be a stable solution—its results will not deteriorate significantly when changing the hyperparameter values.

Some models are ranked differently in terms of mean AUC and PMRA rankings. For example, model XGB5 has a slightly higher mean AUC compared to the RF2 model (0.920 vs. 0.918) while Probability-based Models Ranking Approach classified RF2 considerably higher than XGB5 ($p_{RF2,XGB5}\approx 0.605)$ Wald Chi-square test rejects null hypothesis that models’ performances are the same with *p*-value = 0.04 Comparison of the results models obtained in the specific folds, shown in the Figure 6, leads to the conclusion that in 6 out of 10 of them, RF2 performed better, but at the same time, the XGB5 average AUC is higher. This fact is a consequence of simple averaging approach weaknesses mentioned in Section 2—outliers impact and not including standard deviation differences between folds in analysis. The most significant performance difference in favor of XGB5 occurred in fold 7. It can be considered an outlier because the AUC value obtained in this fold by RF2 significantly deviates from others obtained in most other folds. Moreover, the outstanding performance of RF2 in fold 5 was overshadowed by the lowest standard deviation of results in this fold, as presented in Figure 7. AUC difference is nearly 3 times lower than in the mentioned 7th fold, but performance measured with quantiles, which is presented in Figure 8 is comparable. The above example confirms the Probability-based Models Ranking Approach’s advantages over the classical averaging approach in crucial aspects listed in the Section 2.

### 4.3. Application in Hyperparameter Tuning

After choosing the right algorithm for the problem, the next step is to tune its hyperparameters. The probability-based Models Ranking Approach may also be helpful in this. Parameters *max_depth* and *colsample* of XGBoost [42] algorithm were tuned. The initial grid was:(21)M×C
where:

M={4,8,12,16} (*max_depth* parameter values);

C={0.25,0.5,0.75} (*colsample* parameter values).

In each iteration, the algorithm was trained with the hyperparameters’ sets that were located around the best set and those that the Probability-based Models Ranking Approach recognized as statistically equal to the best one. In each successive iteration, the range from which new sets of parameters were selected decreased twice. The Wald test was used with a 10% significance level to test the significance of the difference between the two models’ performance. The assumed condition to stop further search was that the best model after n+2 iterations is not significantly better than the best model after *n* iterations. The stop condition was fulfilled after the 4th iteration. Figure 9 and Figure 10 present hyperparameters sets which were tested after 2 and 4 iterations. Those recognized as best or not significantly different from best are highlighted on the plot by their shape.

### 4.4. Comparision of Probability-Based Models Ranking Approach and EPP

The EPP and Probability-based Models Ranking Approach were estimated on a credit dataset described in Section 4.1 to check if the obtained probabilities of win and statistical tests *p*-values confirm expected differences between models. Figure 11 and Figure 12 present probabilities of wins estimated in both approaches for the top 10 models. As expected, the differences are small.

The Wald test was performed for every pair from the top 25 models to verify the null hypothesis that their performance is equal. Figure 13 presents obtained *p*-values for both PMRA and EPP, depending on the estimated win probability. It can be observed that for pairs of models whose estimated probability is approximately equal to 0.5, *p*-values obtained from EPP are slightly higher than those from PMRA. Nevertheless, it does not affect model selection decisions because these *p*-values significantly exceed usually used confidence levels. When the win probability is more deviated from 0.5, EPP *p*-values are several times higher than those obtained in the Probability-based Models Ranking Approach. These results are consistent with the theoretical expectations—the EPP model assumes independence of observations, which are, in fact, dependent and therefore underestimates the variance-covariance matrix. It results in the higher *p*-values obtained in statistical tests. Modification introduced in the Probability-based Models Ranking Approach allows for more realistic variances values. It means that the EPP approach may incorrectly reject the null hypothesis, that performance of the two models is equal, while PMRA indicates that there is no reason to reject it. Inference from EPP indications may lead to the false conclusion that it is not worth using a particular algorithm for the problem, even if it was not significantly worse. This unnecessarily limits the range of algorithms available.

## 5. Discussion

The subsampling methods of the model performance assessment have several weaknesses which make their indications insufficient for most practical applications: (i) they are not interpretable (ii) there is no possibility to statistically test the equality of 2 models’ performance (iii) they are not outliers resistant (iv) they do not consider the diversity of different models’ performances on a specific subsample. Our novel Probability-based Models Ranking Approach addressed all these issues. In recent years, many articles have been published showing alternative approaches to the assessment of model performance addressing these issues. As explained in Section 2 the main cause of (i) and (ii) is the bias of the value of the performance metric obtained in the subsampling averaging process. As studies [9,13] prove, bias is particularly large for small-sample problems.

Efforts of some of the newly proposed methods have been focused on reducing the bias of algorithm’s estimated prediction error. Parker et al. show in their study how applying of sample balancing and stratification in methods like *Balanced cv*, *Stratified cv*, *Balanced leave-one-out cv* causes a reduction of averaging bias for small datasets. Another commonly used methods of model performance estimation are *Repeated cross-validation* [43], *Leave-pair-out cross-validation* (LPOCV) [14] and *Leave-one-out cross-validation* (LOOCV) [44]. These methods (especially LPOCV and LOOCV) appear to give the most reliable results possible, but they require fitting the model to data a great many times. This excludes them from applications where the number of calls may be large, such as tuning hyperparameters. The use of these methods is therefore limited to a narrow group of problems, where model fitting time is relatively short. Probability-based Models Ranking Approach’s aim is not to estimate the exact value of the metric of the model’s performance. It is not a necessary feature for every performance assessment procedure (e.g., hyperparameter tuning). Instead, PMRA indicates models’ relative performance based on statistical criteria. The specifics of the time-complexity of PMRA and the methods mentioned above are different. In the case of PMRA, the execution time increases significantly, primarily as the number of models participating in a comparison grows, while for bias reduction methods the factor having the strongest impact on computation time is the number of models’ fits to data.

Other methods attempting to address the weaknesses of classical model performance measures are focused on the statistical aspect of models’ comparison. There is a number of testing procedures determining whether sub-sampling results support the conclusion that the performance of the 2 models differs significantly. Among them *5 × 2-Fold Cross-Validation Paired t-Test* [22] and *Combined 5 × 2 cv F-test* [21] gained the most importance in practical applications. Results provided by these methods may be a valuable decision criterion in the model selection process. Although the above-mentioned methods are reliable, they do not use the full potential of subsampling results. The Probability-based Models Ranking Approach, when used for the creation of models’ ranking or for hyperparameters tuning, compares not only the relative results of 2 analyzed models, but also the results of every other model, fitted to the same subsamples. Thanks to this, PMRA provides a precise and reliable estimate of the probability of win and *p*-value in the statistical test for two models’ performance equality.

## 6. Conclusions and Further Development

In this paper, we identified the most critical weaknesses of subsampling methods, like cross-validation and bagging in the model performance assessment process. Then we presented Elo-based Predictive Power—an innovative performance measure that attempts to respond to the problems of the classical approach. It is interpretable and allows the significance of the difference between 2 models to be tested. Despite its considerable advantages, it has one crucial drawback—*p*-values calculation is based on incorrect assumptions. Further, we introduce the Probability-based Models Ranking Approach—a modified version of EPP. This approach loses EPP’s universality but is based on correct assumptions, and, thanks to that, its indications are reliable.

The basic properties of the Probability-based Model Ranking Approach are its interpretability and the possibility of statistical testing of differences between models’ performance. PMRA may be used to create an interpretable ranking of algorithms to select the one that is most appropriate to the problem. Ranking creation also enables verification of the performance of the newly developed model against a group of state-of-the-art algorithms. PMRA ranking allows us to check whether a new model brings any new quality or its improvement is insignificant.

After the algorithm selection, PMRA may be helpful in the hyperparameters tuning process. It indicates places in the hyperparameters space where it is worth continuing the search and allows to determine the stop conditions reducing the search time to the necessary minimum. PMRA properties may be used to create more complex criteria for tuning continuation depending on the probability of result improvement, statistical tests, and expected computation time. The application of PMRA in developing such criteria is a field for further research. Further research is also needed to examine a reasonable number of folds to apply the PMRA approach and its stability for various numbers of folds.

The results obtained in the study confirm the claims made about the advantages of PMRA over the state-of-the-art averaging approach. However, the results have their limitations. Firstly, the analysis considered multiple machine learning models, but they were trained on a single dataset. This does not negatively influence the validity of findings; however, consideration of a wider range of various datasets would have allowed us to determine in which context PMRA brings the most valuable insight. Secondly, the empirical analysis conducted in Section 4 focused on the qualitative factor. It examined individual cases where the indications of the PMRA measure differed from the state-of-the-art averaging approach and confirmed that the differences were due to the advantages of the new approach over the benchmark subsampling. What was missing, however, was a quantitative study—an examination of how often the two approaches differ in their results—to rank the models differently when creating the models’ ranking, and determine how often parameter tuning leads to similar results in both cases. Conducting the above analysis would have made it possible to determine in which conditions—sample size, number of fitted models—the potentially time-consuming PMRA offers added value over the benchmark approach in model performance assessment.

## Figures and Tables

**Figure 1 sensors-22-06361-f001:**
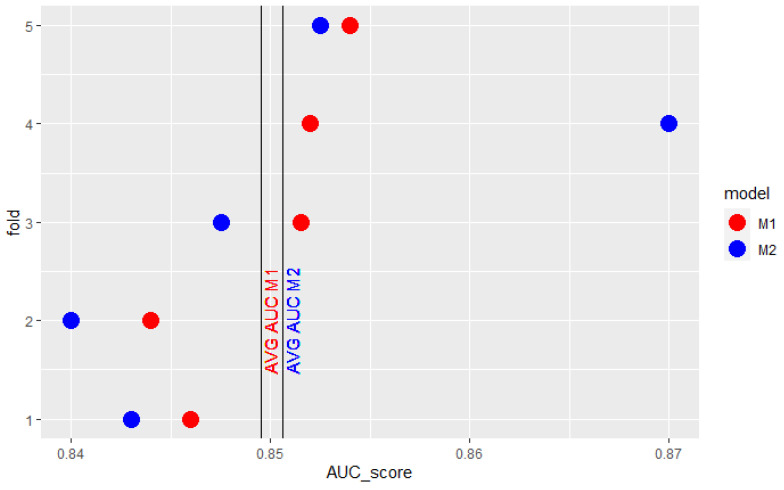
Example AUC scores of 2 models obtained in 5-fold cross-validation.

**Figure 2 sensors-22-06361-f002:**
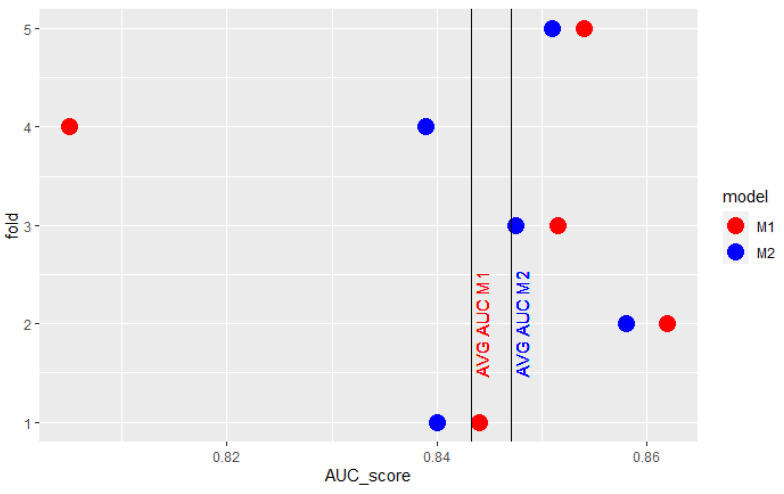
Example AUC scores of 2 models obtained in 5-fold cross-validation.

**Figure 3 sensors-22-06361-f003:**
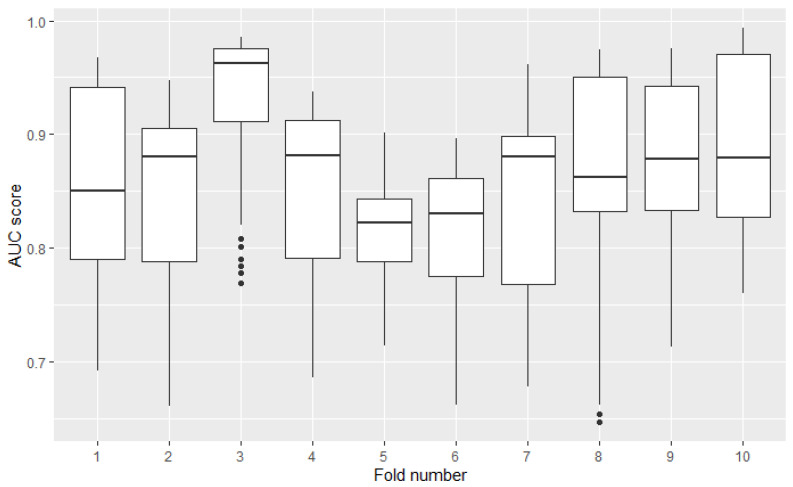
Distribution of 49 models AUC performances among folds obtained from 10-fold cross-validation on credit risk dataset.

**Figure 4 sensors-22-06361-f004:**
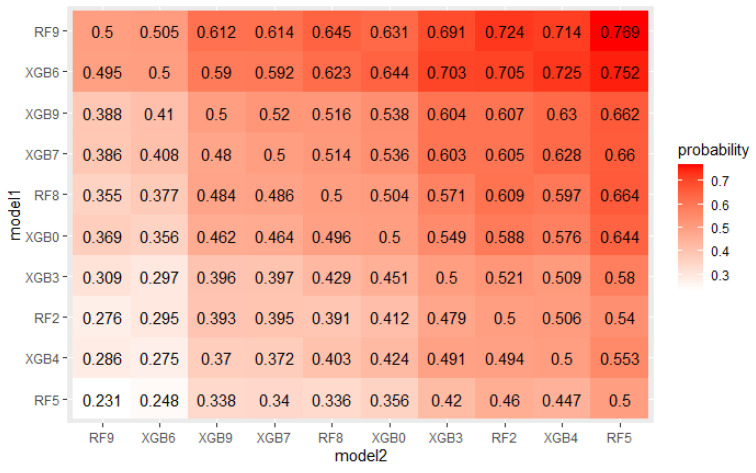
Probabilities of win estimated by PMRA for top 10 models.

**Figure 5 sensors-22-06361-f005:**
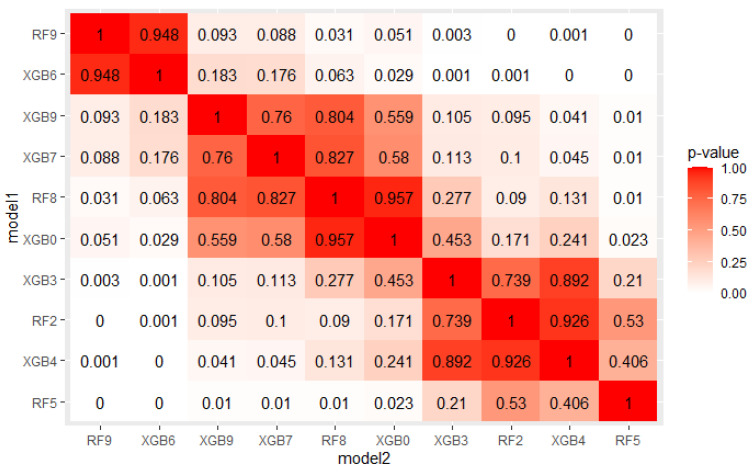
Wald test *p*-value, where null hypothesis is that two models have equal performances.

**Figure 6 sensors-22-06361-f006:**
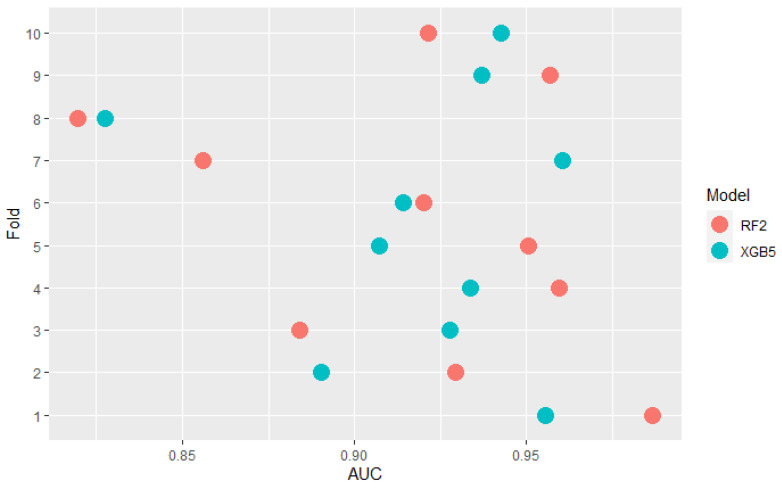
Comparision of RF2 and XGB5 models AUC performance across folds.

**Figure 7 sensors-22-06361-f007:**
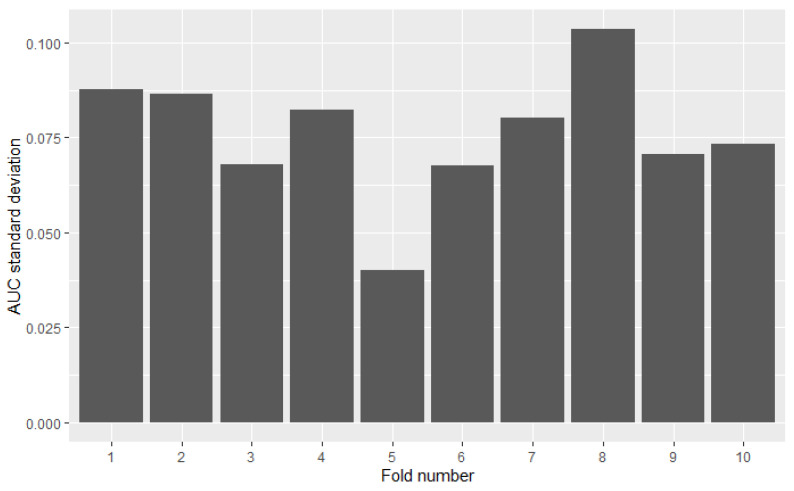
Standard deviation of the models’ performances across folds.

**Figure 8 sensors-22-06361-f008:**
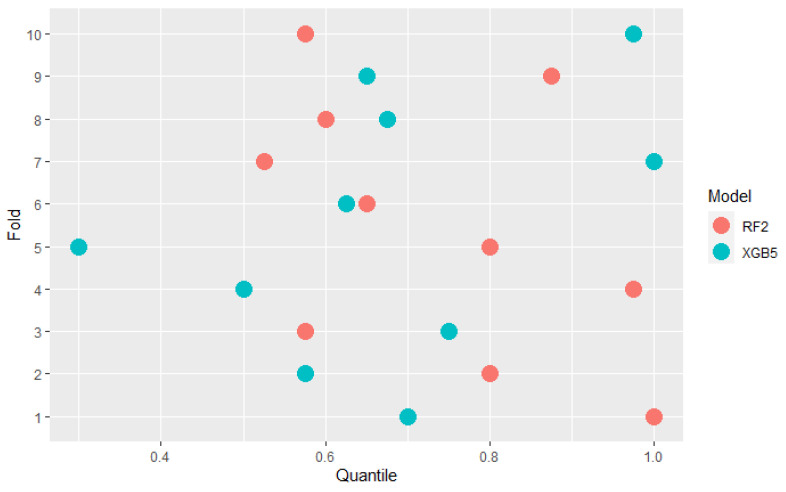
Comparision of RF2 and XGB5 models AUC performance quantiles across folds.

**Figure 9 sensors-22-06361-f009:**
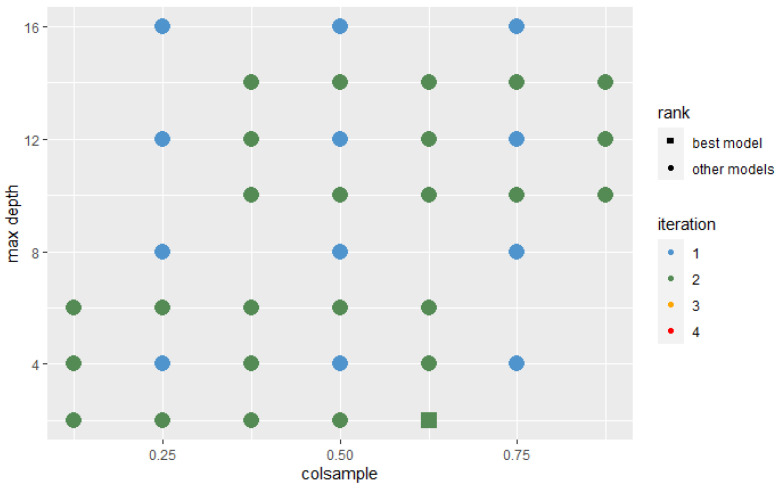
Hyperparameters sets analyzed in 2 iterations.

**Figure 10 sensors-22-06361-f010:**
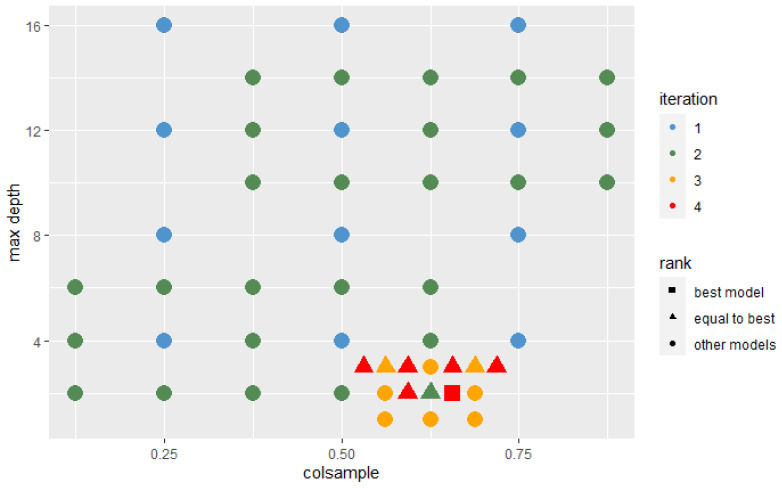
Hyperparameters sets analyzed in 4 iterations.

**Figure 11 sensors-22-06361-f011:**
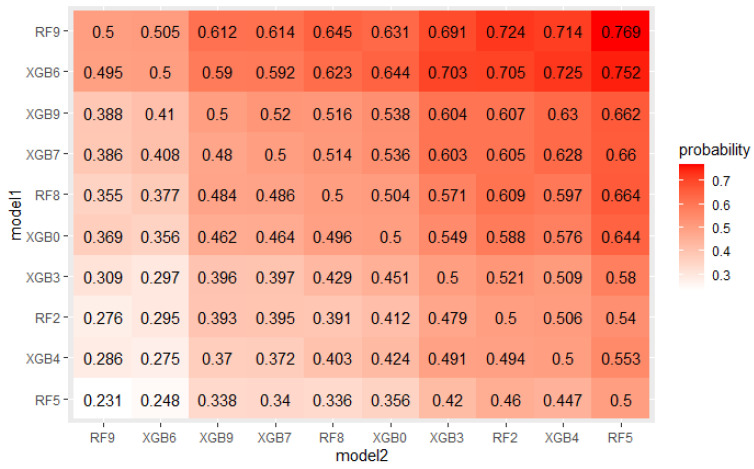
Probabilities of win estimated by PMRA for top 10 models.

**Figure 12 sensors-22-06361-f012:**
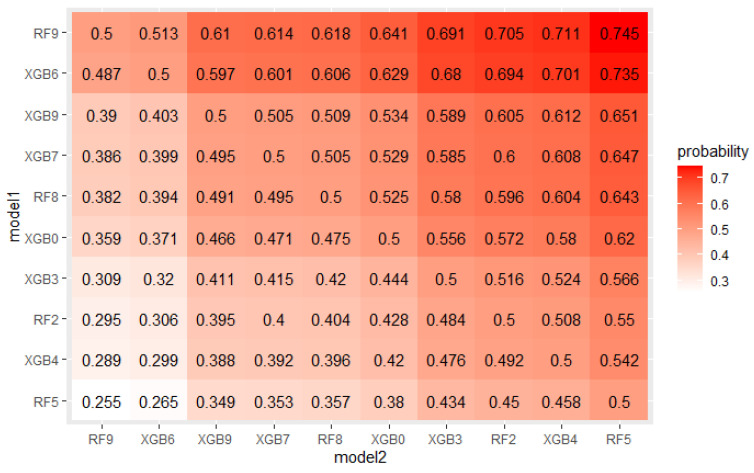
Probabilities of win estimated in EPP approach for the top 10 models.

**Figure 13 sensors-22-06361-f013:**
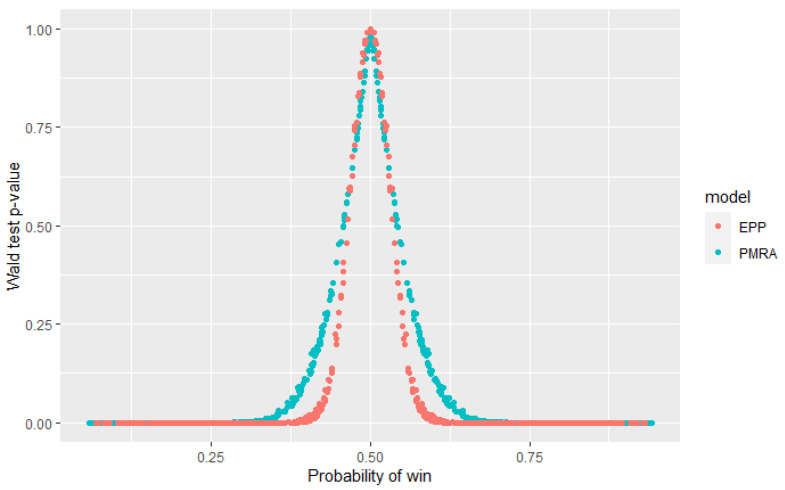
Distribution of the obtained *p*-values in the Wald test according to the probability of winning for PMRA and EPP.

**Table 1 sensors-22-06361-t001:** Example AUC and percentile results of 2 models in 2 folds.

Model	Fold	AUC	Result Percentile
Model 1	1	0.75	60
	2	0.8	75
Model 2	1	0.8	75
	2	0.7	60

**Table 2 sensors-22-06361-t002:** Input dataset.

Model	Fold	AUC
$M_{1}$	1	0.785
$M_{2}$	1	0.743
$M_{3}$	1	0.721
$M_{1}$	2	0.727
$M_{2}$	2	0.672
$M_{3}$	2	0.746

**Table 3 sensors-22-06361-t003:** Transformed dataset.

X1	X2	X3	Fold	Result
1	−1	0	1	1
1	0	−1	1	1
0	1	−1	1	1
1	−1	0	2	1
1	0	−1	2	0
0	1	−1	2	0

**Table 4 sensors-22-06361-t004:** Top categorical variables.

Variable Name	Definition	Levels	Gini
payment_status	Status of the mortgage loan payment	Current 30–59 days delinquent 60–89 days delinquent	0.46
in_mbs	Information whether it belongs to MBS	YES NO	0.20

**Table 5 sensors-22-06361-t005:** Top numeric variables.

Variable Name	Definition	Min	Mean	Max	Gini
business_cycle	Business cycle indicator	2000.17	2012.83	2020.67	0.34
loan_acq_LTV	Unpaid principal balance to the value of all underlying properties.	0	64.81	276.20	0.35
90_days_DELIQ_6 M	How many months in the last 6 M was the credit delinquent at least 90 days	0	0.01	6	0.32
DSCR	A ratio of underwritten net cash flow to an actual debt	0.39	1.78	85.72	0.29
DTB_interest_rate	Difference between reference rates and current credit rates	−2.23	4.39	9.72	0.26
reference_rate_origin	Interbank credits rates at credit origination date	0.07	1.85	6.54	0.25
reference_rate	Reference interest rate at reporting date	0.05	1.10	6.54	0.22
months_on_book	Number of months since credit was granted	0	45.72	236	0.21

**Table 6 sensors-22-06361-t006:** Mean AUC ranking.

	Model	AUC
1	RF9	0.938
2	XGB6	0.933
3	RF8	0.932
4	XGB7	0.931
5	XGB9	0.929
6	XGB0	0.929
7	XGB3	0.926
8	XGB4	0.924
9	RF5	0.920
10	XGB5	0.920

**Table 7 sensors-22-06361-t007:** PMRA ranking.

	Model	P. of Win against Top Model	Wald *p*-Value
1	RF9	-	-
2	XGB6	0.495	0.948
3	XGB9	0.388	0.093
4	XGB7	0.386	0.088
5	RF8	0.355	0.031
6	XGB0	0.369	0.051
7	XGB3	0.309	0.003
8	RF2	0.276	<0.001
9	XGB4	0.286	0.001
10	RF5	0.231	<0.001

## Data Availability

Initial Fannie Mae dataset: https://capitalmarkets.fanniemae.com/credit-risk-transfer/single-family-credit-risk-transfer/fannie-mae-single-family-loan-performance-data; Code and processed data: https://github.com/stagajda/PMRA (accessed on 1 April 2021).

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
