# Peer review of "A Probability-Based Models Ranking Approach: An Alternative Method of Machine-Learning Model Performance Assessment"

_sensors, 2022, doi:10.3390/s22176361_

Round 1

Reviewer 1 Report

This article proposes a new Probability-based Models Ranking Approach. This topic is interesting and its innovation. But it has the following question:

1.The introduction dose not be focused on the research questions, and then note your innovation. The abstract is to long.

2.What is the advantage of the combination of Probability-based Models Ranking Approach comparing other methods. The authors should show an example or some proof.

3. The proposed Probability-based Models Ranking Approach is with objective information. There exists some other Ranking Approach with subjective information, please refer the following works:

A Comprehensive Star Rating Approach for Cruise Ships Based on Interactive Group Decision Making with Personalized Individual Semantics. Journal of Marine Science and Engineering. 2022, 10, 1-20  

A Calibrated Individual Semantic Based Failure Mode and Effect Analysis and Its Application in Industrial Internet Platform. Mathematics 2022, 10, 2492.

Reviewer 2 Report

The paper presents an analysis of theory and empirical research on a Probability-based Ranking Model Approach (PMRA) for models' ranking creation and tuning of hyperparameters. The content of the paper fits within the scope of the Journal. It is a new and original contribution. The paper could be accepted for the publication after minor modifications.

  1. The abstract should be improved and have to provide a more structured aim, scope and background, state the principal objectives and scope of the investigation, describe the methods employed, summarise the results, and state the principal conclusions, recommendations and outlook.
  2. The keywords have to be revised. Please provide some general terms and some subject-specific terms. There should be no less than six and no more than ten keywords. The keywords "Econometrics" and "Performance Metrics" are not mentioned in the paper.
  3. The aim of the paper should be clearly stated in one sentence and presented in the introduction.
  4. The theoretical study should include more recently published papers from high-level scientific journals indexed in Clarivate Analytics Web of Science and Scopus databases (years 2019-2022).
  5. Comparisons with other studies have to be provided in the discussion section. Please interpret and describe the significance of your findings in light of what was already known about the research problem being investigated and explain any new understanding or fresh insights about the problem after you've taken the findings into consideration. Please provide a comparison with other studies.
  6. Please briefly describe in the last paragraph of the INTRODUCTION section the content of each section of the paper and include brief information on methods (one sentence).
  7. Please provide research LIMITATIONS and identify potential weaknesses. Comment on their relative importance in relation to your overall interpretation of the results and, if necessary, note how they may affect the validity of the findings.
